# The Association of Metformin, Other Antidiabetic Medications, and Statins with the Prognosis of Hepatocellular Carcinoma in Patients with Type 2 Diabetes: A Retrospective Cohort Study

**DOI:** 10.3390/biomedicines12081654

**Published:** 2024-07-24

**Authors:** Iida Tuunanen, Ari Hautakoski, Heikki Huhtamäki, Martti Arffman, Reijo Sund, Ulla Puistola, Peeter Karihtala, Arja Jukkola, Elina Urpilainen

**Affiliations:** 1Department of Obstetrics and Gynecology, Oulu University Hospital, Wellbeing Services County of North Ostrobothnia, University of Oulu, 90220 Oulu, Finland; 2Research Unit of Clinical Medicine, University of Oulu, 90220 Oulu, Finland; 3Medical Research Center, Oulu University Hospital, Wellbeing Services County of North Ostrobothnia, University of Oulu, 90220 Oulu, Finland; 4Research Unit of Mathematical Sciences, Faculty of Science, University of Oulu, 90014 Oulu, Finland; 5Department of Public Health and Welfare, Finnish Institute for Health and Welfare, 00271 Helsinki, Finland; 6Institute of Clinical Medicine, University of Eastern Finland, 70029 Kuopio, Finland; 7Department of Oncology, University of Helsinki, Helsinki University Hospital Comprehensive Cancer Center, 00029 Helsinki, Finland; 8Department of Oncology and Radiotherapy, Tays Cancer Center, Tampere University Hospital, Faculty of Medicine and Health Technology, Tampere University, 33521 Tampere, Finland; arja.jukkola@tuni.fi

**Keywords:** diabetes mellitus, epidemiology, liver neoplasms, malignancy, mortality

## Abstract

This study aimed to explore whether the prediagnostic use of metformin and statins is associated with the prognosis of patients with hepatocellular carcinoma (HCC) and type 2 diabetes. We identified 1383 eligible individuals who had both type 2 diabetes and HCC diagnosed between 1998 and 2017 from several Finnish registers. Cox models were fitted for cause-specific and all-cause mortality in relation to the use of antidiabetic medications and statins prior to the HCC diagnosis. Prediagnostic metformin use was associated with decreased overall mortality (hazard ratio 0.84, 95% confidence interval 0.74–0.94) compared with nonuse in patients with type 2 diabetes. Similarly, slightly decreased HCC mortality and other-cause mortality were observed among metformin users. The results were inconclusive regarding metformin use and both overall and HCC mortality among patients with localized HCC. No discernible contrast between statin users and nonusers was found in overall mortality nor HCC mortality in either the whole cohort or patients with localized cancer.

## 1. Introduction

Liver cancer is the sixth most commonly diagnosed cancer worldwide, with 866,136 new cases reported in 2022 [1]. It ranked as the third-highest cause of cancer deaths in the same year with 758,725 deaths [1]. According to the Finnish Cancer Registry, there were 513 new cases of liver cancer in Finland in 2022 [2]. The most common type of primary liver cancer is hepatocellular carcinoma (HCC), amounting to approximately 90% of all cases [3]. A major risk factor for HCC is perpetual infection with the hepatitis B or C virus and alcohol abuse [4,5]. Particularly in developed countries, nonalcoholic fatty liver disease and nonalcoholic steatohepatitis may soon replace alcohol- and viral-related liver diseases as the major risk factors for HCC. Nonalcoholic fatty liver disease develops due to metabolic syndrome, which is strongly associated with obesity, dyslipidemia, hypertension, and type 2 diabetes (T2D) [6].

HCC patients with T2D have a lower overall survival (OS) rate and increased mortality compared to HCC patients without T2D [7,8,9,10]. Some recent meta-analyses indicate that first-line T2D treatment with metformin is associated with prolonged OS after curative treatments for HCC in both Caucasian and Asian populations [11,12].

People with T2D are often administered statins as the primary prevention of cardiovascular disease and due to diabetes-associated cardiovascular disease [13]. In Finland, 46% of the patients diagnosed with diabetes use lipid-lowering medication within 6–12 months after their T2D diagnosis [14]. Statin use is associated with a reduced risk of HCC, reduced all-cause mortality in the patients with HCC, decreased recurrence of HCC, and reduced incidence of HCC in people with T2D [15,16,17]. The effect of statin use on the HCC-specific survival of people with T2D has not been widely investigated, but in Antwi et al.’s [18] study, neither pre- nor postdiagnostic statin use had an association with the OS of HCC patients.

As the previous results on this topic have been variable and mostly lacked information on the drug dose and consumption time, this study aimed to evaluate whether the prediagnostic use of metformin, other antidiabetic medications (ADMs), or statins are associated with better overall and cause-specific survival in HCC patients with T2D by using a robust study design and data from multiple high-quality Finnish national registers.

## 2. Materials and Methods

### 2.1. Data Sources

In this study, we complied with the Strengthening the Reporting of Observational Studies in Epidemiology guidelines [19]. The data source comprising persons diagnosed with diabetes was the “Diabetes in Finland” (FinDM) database, which was originally generated to make the nationwide epidemiological monitoring of diabetes in Finland possible [20]. Data from numerous registers are included: the Special Reimbursement Register and the Prescription Register from the Social Insurance Institution of Finland, the Care Register for Health and the Hospital Discharge Register from the Finnish Institute for Health and Welfare, and the cause of death statistics from Statistics Finland (Appendix A). The Prescription Register provides systematic records of the drugs purchased since 1994, as it incorporates data about the medications prescribed by doctors and reimbursed by the Social Insurance Institution of Finland. In the FinDM database, the onset of diabetes is defined as the first entry regarding diabetes in the aforementioned registers. Individuals with diabetes were classified into type 1 and type 2 based on information on their diagnoses, medication, and age in the first year following the recorded onset of diabetes. The database shows good coverage against a local register in Southern Finland, according to data reliability studies [21].

Data on HCC cases were acquired from the Finnish Cancer Registry, which is linked to the FinDM database. The Finnish Cancer Registry incorporates information on the cancer cases diagnosed since 1953 in Finland, including the date of diagnosis, morphology, histology, and stage [22]. Follow-up data, including causes of death and dates, were obtained from the cause of death statistics, from which the coding experts of the Finnish Cancer Registry further determined the cancer specificity by examining the cause of death records together with other data on the cancer in question.

### 2.2. Study Population

The cohort formation process is presented in the flowchart below (Figure 1). We identified 2402 people with T2D and HCC. The individuals with T2D who were diagnosed before their 40th birthday were excluded. Furthermore, we excluded the individuals with HCC who were diagnosed before 1998, those with other previous cancer diagnoses (except nonmelanoma skin cancer), and the HCC cases that were diagnosed at autopsy. In addition, those individuals with T2D who were diagnosed less than 365 days before their HCC diagnosis were excluded to avoid reverse causality and immortal time bias as well as to enable the collection of sufficient medication data from before the HCC diagnosis. Follow-up began on the date of the HCC diagnosis. The final cohort consisted of 1383 people who had both type 2 diabetes and HCC diagnosed between 1998 and 2017.

The education levels of the study participants were divided into three categories (basic, intermediate, and higher) based on the most advanced grade or degree achieved by an individual. Basic education refers to a person who has graduated only from obligatory comprehensive schooling, which compromises nine grades. Intermediate education refers to a person who has graduated from upper secondary education, which includes three years of study in a general upper secondary school or in vocational education. Higher education refers to a person who has graduated from a university or a university of applied sciences.

The presence of hepatitis or cirrhosis were identified based on register data which were coded using the *International Classification of Diseases* versions 8, 9, and 10 (Appendix A).

### 2.3. Classification of Medication

ADM use in the cohort was categorized into (1) metformin users and nonusers, (2) insulin users and nonusers, and (3) other ADM users and nonusers. The cohort was also categorized into statin users and nonusers, regardless of the ADM type used. The criterion for statin, metformin, insulin, and other ADM use was the purchase of the medication for a minimum of 180 days within three years before the HCC diagnosis, which guarantees a continuous use of medication. A nonuser was defined as making a statin or an ADM purchase less than 180 days prior to their diagnosis, or not all. A person using multiple medications could be categorized into more than one medication user group. Cumulative medication use was considered by determining the defined daily doses (DDDs) for three years prior to the HCC diagnosis. HCC was defined using the diagnosis code C22.0 of the *International Statistical Classification of Diseases and Related Health Problems* (10th revision) and the *International Classification of Diseases for Oncology* (third edition), as well as the morphology code M8170/3 (HCC, not otherwise specified).

### 2.4. Staging

The HCC stage was defined based on the Finnish Cancer Registry classification. The stage of HCC at diagnosis is categorized in the Finnish Cancer Registry as follows: (0) unknown; (1) localized; (2) nonlocalized, only regional lymph node metastases; (3) metastasized or an invasion of adjacent tissues; (4) nonlocalized, no information on extent; (5) locally advanced, an invasion of adjacent tissues; and (6) nonlocalized, also distant lymph node metastases. Based on these data, we categorized the stages in our study as local (1), advanced (2–6), and unknown (0).

### 2.5. Statistical Methods

The follow-up started on the date of the HCC diagnosis and ended on the date of death, emigration, or closure of the follow-up on 31 December 2017. The cumulative mortality from HCC and from other causes of death in the prediagnostic ADM and statin groups was calculated using the Aalen–Johansen estimator of the cumulative incidence function for competing risks. Cause-specific mortality rates were analyzed using Cox proportional hazard models to obtain estimated hazard ratios (HRs) with 95% confidence intervals (CIs), adjusting for the confounding effects of the patients’ age, year of diagnosis, duration of diabetes, and HCC stage. A possible nonlinear dose-dependent effect of the medications was assessed by replacing the medication group indicators in the Cox models with cubic spline terms for the average amount of DDDs per day in different medication groups. These models were fitted for patients with all stages of HCC as well as for patients with localized HCC.

The data obtained from the registers were prepared and analyzed using SAS (version 9.4) and R (version 4.0.5). All of the statistical analyses were performed in the R environment. The functions in the “survival” package of R functions were used to compute the Aalen–Johansen estimators of cumulative mortality by cause to fit the Cox models and to diagnose possible deviations from the assumptions of the underlying model. The Schoenfield residuals against the transformed time as well as the cloglog figures were visually inspected to verify that the proportionality assumptions of the Cox regression model were met. Missing data were encountered only concerning the HCC stage, and we labeled these cases as “unknown”.

## 3. Results

### 3.1. Analysis of the Whole Cohort

A total of 1383 HCC cases were collected, with a median follow-up time of 0.4 years and an interquartile range of 0.08 to 1.3 years (Table 1). Most of the HCC cases were diagnosed when patients were between the ages of 65 and 74 years, with a median age of 72 years (Appendix A). The majority of the study population was male (*n* = 1109, 80%). Two hundred and ninety-two patients (21%) were recorded as local and three hundred and sixty-three (26%) as advanced. In the remaining 728 patients (53%), the stage was unknown.

When comparing the 885 metformin users to the 498 nonusers, the former had a longer diabetes duration at HCC diagnosis, with a median of 9.9 years compared to 9.0 years in the nonusers (Table 1). The most commonly used other ADMs were sulfonylureas (71%) and dipeptidyl peptidase-4 inhibitors (29%) (Appendix A). Within the metformin user group, 35% of patients also used insulin; in the metformin nonuser group, 44% used insulin. Similarly, within the metformin user group, 44% used statins, and among metformin nonusers, 27% used statins.

The 529 statin users in the study cohort had a longer median diabetes duration (11.4 years) than statin nonusers (8.6 years) (Table 1). The most commonly used statins were simvastatin (72%) and atorvastatin (23%) (Appendix A).

During follow-up, 1210 deaths occurred in the cohort, with 1025 deaths due to HCC. Through the analysis of all the stages, evidence was found for an association with decreased overall mortality among prediagnostic metformin users (HR 0.84, 95% CI 0.74–0.94) compared to metformin nonusers. Similarly, slightly decreased HCC mortality (HR 0.86, 95% CI 0.75–0.99) and other-cause mortality (HR 0.71 95% CI 0.53–0.97) were observed among metformin users. For both overall mortality and HCC mortality, the results were inconclusive for insulin use and other ADM use (Table 2). The estimation results were adjusted for the patients’ sex, level of education, duration of diabetes, and presence of hepatitis and cirrhosis, as well as for the year of and age at the HCC diagnosis and the HCC stage (Appendix A).

The five-year cumulative overall mortality was 93% in metformin users and 95% in nonusers (Figure 2A). Regarding HCC-specific mortality, these proportions were 81% and 77% in metformin users and nonusers, respectively (Figure 2B). The five-year cumulative mortality from other causes was 12% for metformin users and 18% for nonusers (Figure 2C).

No discernible contrast between statin users and nonusers was found in either overall mortality (HR 0.91, 95% CI 0.80–1.04) nor HCC mortality (HR 0.91, 95% CI 0.80–1.05) (Table 2).

The five-year cumulative overall mortality was 94% in the statin user group as well as in the nonuser group (Figure 2D). Furthermore, no significant differences were found in cumulative HCC mortality or mortality from other causes (Figure 2E,F). The DDD analysis indicated no association between the cumulative amount of any medication used and overall (Appendix A) nor HCC mortality.

### 3.2. Analysis of the Patients with Localized HCC

In the subgroup of patients with localized HCC (*n* = 292) the median follow-up time was 0.7 years, and the interquartile range was 0.1 to 2.7 years (Table 3). Most of the local HCC cases were diagnosed when patients were between the ages of 65 and 74 years, with a median age of 71 years (Appendix A).

There were 173 metformin users in the local HCC group, with a median diabetes duration of 9.6 years. Metformin users had a median follow-up time of 0.8 years. When comparing the metformin users with the nonusers, metformin nonusers had a shorter follow-up time of 0.6 years, but between the two groups there were no remarkable differences in diabetes duration (Table 3).

Statin users with localized HCC (*n* = 97) had a median diabetes duration of 10.2 years and a median follow-up time of 0.6 years. In the 195 statin nonusers, the median diabetes duration was 9.3 years and the median follow-up time 0.7 years (Table 3).

During the follow-up of patients with localized HCC, 269 deaths occurred, with 207 deaths due to HCC. In this subgroup, the results were inconclusive due to the small number of cases between metformin use and overall mortality (HR 0.96, 95% CI 0.74–1.24), as well as metformin use and HCC mortality (HR 1.03, 95% CI 0.77–1.39) (Table 4). The estimation results were adjusted for the patients’ sex, level of education, duration of diabetes, and presence of hepatitis and cirrhosis, as well as the year of and age at the HCC diagnosis and the HCC stage (Appendix A).

In the local subgroup, the five-year cumulative overall mortality was 89% in metformin users and 91% in nonusers (Figure 3A). The cumulative HCC mortality was 70% in metformin users and 68% in nonusers (Figure 3B), and the cumulative mortality from other causes was 19% in metformin users and 23% in nonusers (Figure 3C).

The results were inconclusive in terms of statin use and both overall mortality (HR 0.92, 95% CI 0.69–1.23) and HCC mortality (HR 0.87, 95% CI 0.62–1.20) in the local subgroup (Table 4). The estimation results were adjusted for the patients’ sex, level of education, duration of diabetes, and presence of hepatitis and cirrhosis, as well as the year of and age at the HCC diagnosis and the HCC stage (Appendix A).

In statin users with localized stage HCC, the five-year cumulative overall mortality was 92%, and 89% in nonusers (Figure 3D). The cumulative HCC mortality was 68% in statin users and 70% in nonusers (Figure 3E). Cumulative mortality from other causes was 24% in statin users and 19% in nonusers (Figure 3F). The DDD analysis indicated no association between the cumulative amount of any medication used and the prognosis of localized HCC (Appendix A).

## 4. Discussion

In this study, prediagnostic metformin use seemed to have an association with decreased overall mortality in patients with HCC and T2D. However, in a subgroup of patients with localized HCC, the results were inconclusive between metformin use and both overall and HCC mortality. In terms of prediagnostic statin use, no discernible contrast between users and nonusers was found in either group in terms of overall mortality nor HCC mortality.

The mechanisms underlying the possible antitumor effects of metformin are of great interest. Through direct activation of adenosine monophosphate-activated protein kinase, metformin can activate p53 and thus stop the cell cycle (Figure 4). In addition, the indirect activation of this protein kinase also decreases the activity of mTOR, which has an influence on protein synthesis and cell growth [23]. In HCC, metformin has been reported to inhibit invasion and growth and activate apoptosis and autophagy [24].

According to our analysis of the whole cohort and all of the HCC stages, metformin use seemed to be associated with decreased overall mortality and thus prolonged OS. This finding is in line with a previous meta-analysis in which, with limited knowledge of the HCC stage, metformin use was associated with decreased all-cause mortality and recurrence-free survival in patients with HCC [11]. However, the analysis did not consist of study populations of patients with diabetes only, and there was limited information on the metformin dose and consumption time. According to another meta-analysis of eight retrospective cohort studies, the survival of noncuratively treated advanced-stage HCC patients with T2D was not more strongly associated with metformin use compared to the use of other ADMs [12].

In our study, in a subgroup of patients with localized HCC, the results were inconclusive between metformin use and both overall and HCC mortality. However, in a meta-analysis, Zhou et al. [12] found that after curative treatment of HCC, metformin use had an association with longer OS and recurrence-free survival compared to the use of other ADMs. However, this meta-analysis did not comprise study populations of patients with diabetes alone. Another retrospective cohort study also found that metformin use was associated with decreased HCC-specific mortality after curative HCC treatment in patients with T2D [27]. In addition, in a systematic review, metformin use was associated with prolonged OS and a decreased HCC recurrence rate in patients with T2D after curative HCC therapy [28].

Regarding statin use, an experimental study in a HCC model reported that statins might prolong the survival of animals with HCC [29]. The underlying mechanisms may be linked to statins activating G1/S cell cycle arrest and inducing apoptosis [30], as well as inhibiting the P13K-AKT-mTOR pathway, which is commonly activated in HCC [31] (Figure 5).

In terms of prediagnostic statin use, no discernible contrast between users and nonusers was found in the overall mortality nor HCC mortality in patients with T2D. Therefore, this finding is in line with a previous study that found no association between pre- and postdiagnostic statin use and the OS of elderly HCC patients with T2D [18]. According to our knowledge, the association between statin use and HCC survival in people with T2D has not been widely investigated. In a recent meta-analysis of HCC patients in which diabetes was not taken into account, statin use was associated with decreases in all-cause mortality in HCC patients, HCC-related mortality, and in addition, HCC recurrence after curative treatments [15]. Another recent meta-analysis of HCC patients in general demonstrated an association between statin use and decreased all-cause mortality, but not HCC-related mortality [16].

In our study, the DDD analysis indicated no association with the HCC prognosis. A previous retrospective cohort study found an association between the prediagnostic use of metformin with a dose ≤ 1500 mg/day and a decreased risk of death [18]. A null association with OS was found for prediagnostic metformin use with a dose > 1500 mg/day and for postdiagnostic metformin use. However, the study was based only on the data of patients aged 65 years and older [18].

The strengths of our study are the high-quality registers used and the study design, which has a minimal risk of biases. The data in the Finnish registers, such as the Hospital Discharge Register and the Finnish Cancer Registry, are reliable, and the registers are comprehensive [33,34,35]. Furthermore, the coverage in the Prescription Register of reimbursed medications prescribed by physicians is virtually complete for the study period [36]. We also had detailed data on medication use since our criterion required at least six months of medication use for ADMs in the three years prior to the HCC diagnosis. Data on DDDs were available, and adjustments for the patients’ age, sex, level of education, diabetes duration, follow-up time, and the presence of cirrhosis and hepatitis were performed. The study cohort considered virtually all cases of HCC and individuals with T2D in Finland during the study period.

The main weakness of this study is that we only had register information available, with no comprehensive information about the HCC treatments. The registers used did not take lifestyle parameters into account. Furthermore, the degrees of hypercholesterinemia, atherosclerosis, hypertension, and left ventricular function are not available in the register data. Also, the duration of cardiovascular diseases is not available. The laboratory values and severity of diabetes are not known, but insulin use and the duration of diabetes are considered to be surrogate markers for the stage of diabetes. In addition, in a high proportion of HCC cases the stage was recorded as unknown (over 50%). Although the clinical staging of HCC might be challenging, it is highly likely that most unknown cases were advanced. In addition, the median follow-up time in both analyses might seem short, but the five-year survival of the subgroup complied with the general survival time of patients with HCC. The short follow-up time was affected by many patients dying during the first month of the follow-up. In the first two weeks, 165 patients died from HCC. Furthermore, within the first two weeks, 74 patients died from causes other than HCC; the majority of these patients died from cardiovascular related causes (42%) or gastrointestinal incidents (38%), including hepatic cirrhosis.

Challenges of confounding by indication are present in observational studies which contain endpoints that have not yet been studied in randomized controlled trials [37]. As various types of medication are initiated to treat conditions other than the one in the focus of an observational study, differences in participants can have an impact on the results. For example, patients with T2D have worse HCC prognoses compared to patients without T2D [38], but hypercholesterolemia seems not to decrease the survival of HCC patients [39]. Much has been discussed about either metformin or statin use association with healthy user bias, as, for example, statin use is linked to healthy lifestyles, adherence to treatments, better tolerance of side-effects, and the absence of contraindications [40]. However, statin users appear more likely to be elderly and have more coincident severe cardiovascular comorbidities than nonusers of statins [41]. In our study, metformin users had only slightly more insulin, other oral ADM, and statin use compared to metformin nonusers. The prognostic effect of metformin on overall mortality can be partly caused by the reduction of cardiovascular risk factors. However, the prediagnostic metformin use was associated with slightly decreased HCC-specific mortality, which is likely explained by metformin’s cardiovascular effects.

In conclusion, prediagnostic metformin use was associated with decreased overall mortality, as well as slightly decreased HCC mortality and other-cause mortality in patients with T2D. In terms of localized HCC, results were inconclusive between prediagnostic metformin use and mortality in patients with HCC. Similarly, no discernible contrast between prediagnostic statin users and nonusers was found in overall mortality nor HCC mortality in either the whole cohort or the local subgroup.

## Figures and Tables

**Figure 1 biomedicines-12-01654-f001:**
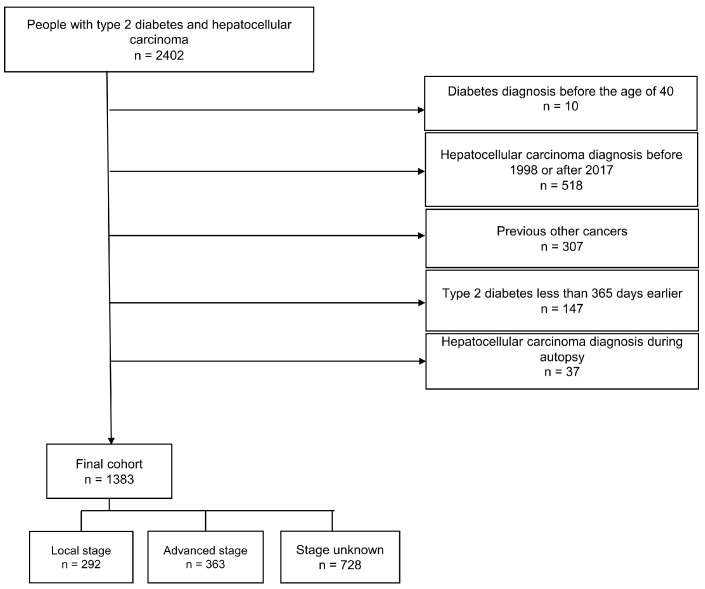
Flowchart of the cohort selection process.

**Figure 2 biomedicines-12-01654-f002:**
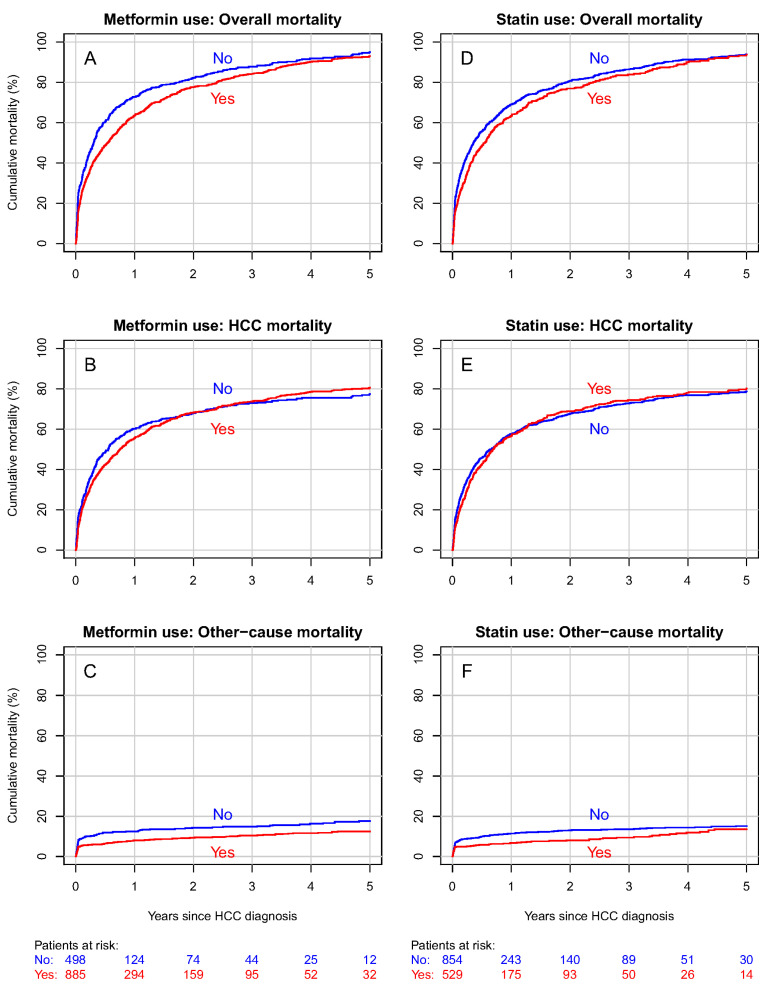
Cumulative overall, hepatocellular carcinoma (HCC) and other-cause mortality curves for the whole cohort according to metformin (**A**–**C**) and statin (**D**–**F**) use. Curves (**A**,**D**) represent overall mortality, curves (**B**) and E HCC mortality, and curves (**C**,**F**) other-cause mortality.

**Figure 3 biomedicines-12-01654-f003:**
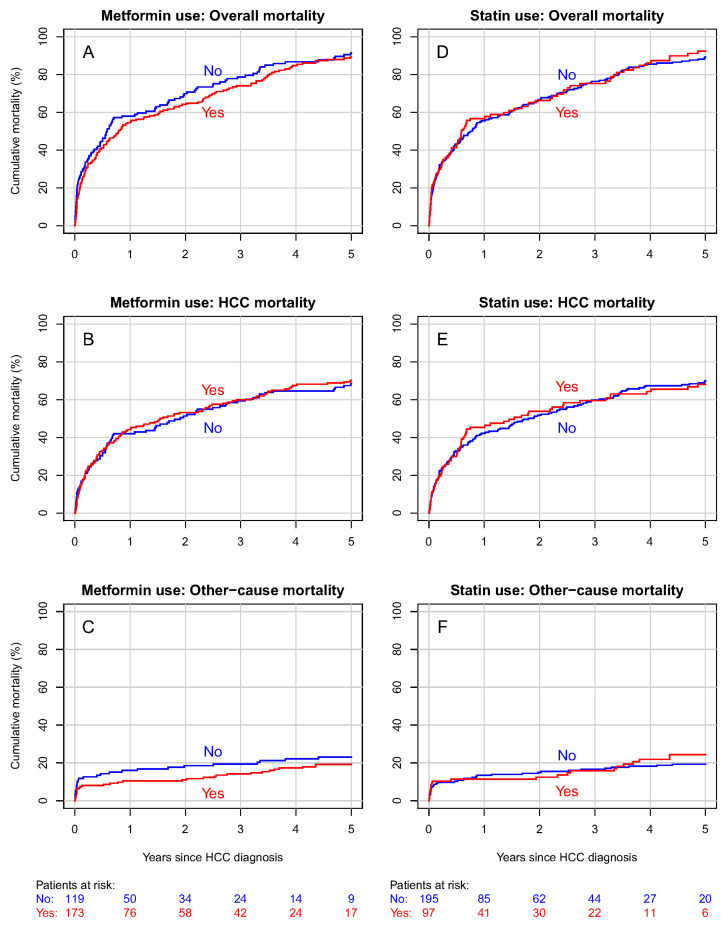
Cumulative overall, HCC, and other-cause mortality curves for the subgroup of patients with local HCC according to metformin (**A**–**C**) and statin (**D**–**F**) use. Curves (**A**,**D**) represent overall mortality, curves (**B**) and E HCC mortality, and curves (**C**,**F**) other-cause mortality.

**Figure 4 biomedicines-12-01654-f004:**
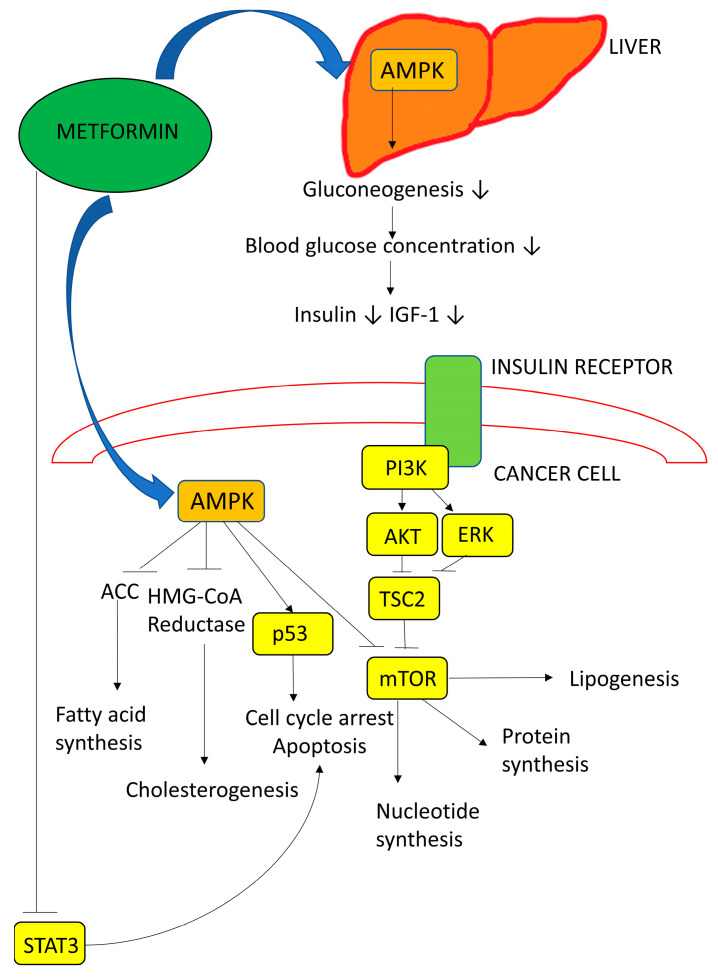
Metformin has both direct and indirect effects on cancer cells. It activates AMP-activated protein kinase (AMPK), leading to the inhibition of the mammalian target of rapamycin (mTOR). It also sensitizes tissues to insulin, reduces hepatic gluconeogenesis, and decreases circulating insulin levels. This leads indirectly to reduced phosphatidylinositol-3-kinase (PI3K) signaling. IGF-1 = insulin-like growth factor 1, ACC = acetyl-CoA carboxylase, HMG-CoA = 3-hydroxy-3-methyl-glutarylcoenzyme A, p53 = tumor protein p53, AKT = serine/threonine-specific protein kinase, and STAT3 = signal transducer and activator of transcription 3, ↓ decreasing effect, ⊥ inhibitory effect [25,26].

**Figure 5 biomedicines-12-01654-f005:**
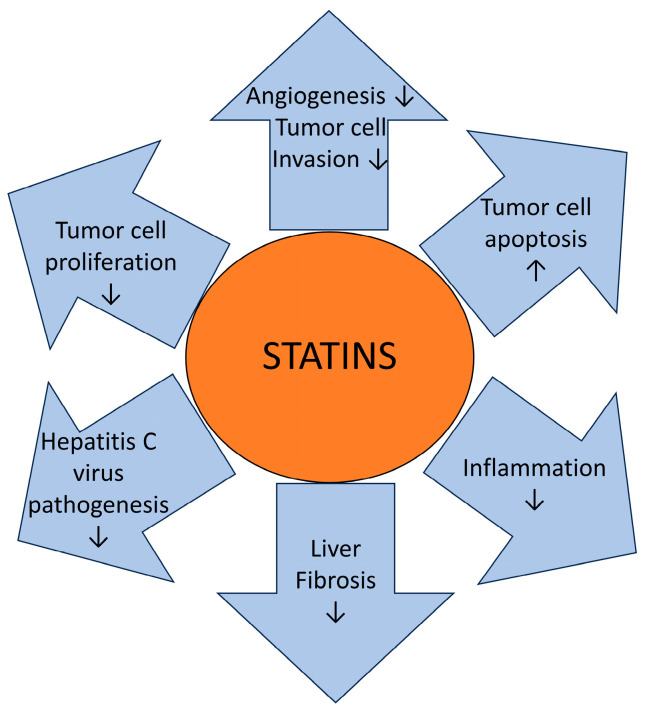
Statins may have multiple antitumoral mechanisms in hepatocellular carcinoma. Statins seems to increase tumor cell apoptosis as well as decrease both tumor cell invasion and proliferation. Additionally, statins decrease angiogenesis, inflammation, liver fibrosis, and hepatitis C virus pathogenesis. ↓ decreasing effect, ↑ increasing effect [32].

**Table 1 biomedicines-12-01654-t001:** Patient characteristics of the whole cohort according to antidiabetic medication and statin use.

	Metformin		Insulin		Other ADM		Statin		
	Yes (%)	No (%)	Yes (%)	No (%)	Yes (%)	No (%)	Yes (%)	No (%)	Total (%)
**Total**	885 (100)	498 (100)	533 (100)	850 (100)	749 (100)	634 (100)	529 (100)	854 (100)	1383 (100)
**Sex**									
Male	733 (83)	376 (76)	426 (80)	683 (80)	598 (80)	511 (81)	435 (82)	674 (79)	1109 (80)
Female	152 (17)	122 (24)	107 (20)	167 (20)	151 (20)	123 (19)	94 (18)	180 (21)	274 (20)
**Education**									
Basic	275 (31)	149 (30)	178 (33)	246 (29)	235 (31)	189 (30)	168 (32)	256 (30)	424 (31)
Intermediate	338 (38)	207 (42)	206 (39)	339 (40)	281 (38)	264 (42)	213 (40)	332 (39)	545 (39)
Higher	272 (31)	142 (29)	149 (28)	265 (31)	233 (31)	181 (29)	148 (28)	266 (31)	414 (30)
**Follow-up time (years)**									
Median	0.5	0.3	0.4	0.4	0.4	0.4	0.5	0.3	0.4
Interquartile range	0.1–1.5	0.04–1.0	0.07–1.2	0.09–1.3	0.08–1.3	0.09–1.2	0.1–1.4	0.06–1.2	0.08–1.3
**Duration of diabetes (years)**									
Median	9.9	9.0	13.7	7.7	10.4	8.5	11.4	8.6	9.6
Interquartile range	6.1–14.9	5.1–15.9	9.1–18.1	4.4–11.7	6.9–15.2	4.3–15.2	7.2–16.7	5.0–13.5	5.6–15.2
**Hepatitis**									
Yes	55 (6)	48 (10)	47 (9)	56 (7)	50 (7)	53 (8)	34 (6)	69 (8)	103 (7)
No	830 (94)	450 (90)	486 (91)	794 (93)	699 (93)	581 (92)	495 (94)	785 (92)	1280 (93)
**Cirrhosis**									
Yes	190 (21)	144 (29)	152 (29)	182 (21)	170 (23)	164 (26)	91 (17)	243 (28)	334 (24)
No	695 (79)	354 (71)	381 (71)	668 (79)	579 (77)	470 (74)	438 (83)	611 (72)	1049 (76)
**Age at HCC diagnosis (years)**									
Median	71	72	71	72	72	71	72	71	72
Interquartile range	66–78	65–80	65–77	66–79	66–79	65–78	67–78	65–78	66–78
**Stage of HCC**									
Local	173 (20)	119 (24)	105 (20)	187 (22)	167 (22)	125 (20)	97 (18)	195 (23)	292 (21)
Advanced	228 (26)	135 (27)	153 (29)	210 (25)	204 (27)	159 (25)	122 (23)	241 (28)	363 (26)
Unknown	484 (55)	244 (49)	275 (52)	453 (53)	378 (50)	350 (55)	310 (59)	418 (49)	728 (53)

ADM = antidiabetic medication, HCC = hepatocellular carcinoma.

**Table 2 biomedicines-12-01654-t002:** Estimation results from Cox proportional hazard models of mortality from all causes, HCC, and other causes in the whole cohort, adjusted for sex, education, duration of diabetes, and the presence of hepatitis and cirrhosis, as well as year of and age at HCC diagnosis and HCC stage.

	Overall Mortality			HCC Mortality			Other-Cause Mortality		
	HR	95% CI	*n*	HR	95% CI	*n*	HR	95% CI	*n*
**Prediagnostic metformin use**									
Yes	0.84	(0.74–0.94)	762	0.86	(0.75–0.99)	660	0.71	(0.53–0.97)	102
No	1.00	Ref.	448	1.00	Ref.	365	1.00	Ref.	83
**Prediagnostic other ADM use**									
Yes	0.99	(0.88–1.11)	664	0.97	(0.85–1.11)	558	1.04	(0.76–1.42)	106
No	1.00	Ref.	546	1.00	Ref.	467	1.00	Ref.	79
**Prediagnostic insulin use**									
Yes	1.00	(0.88–1.14)	468	0.99	(0.86–1.14)	397	1.09	(0.78–1.52)	71
No	1.00	Ref.	742	1.00	Ref.	628	1.00	Ref.	114
**Prediagnostic statin use**									
Yes	0.91	(0.80–1.04)	444	0.91	(0.80–1.05)	384	0.88	(0.62–1.23)	60
No	1.00	Ref.	766	1.00	Ref.	641	1.00	Ref.	125

ADM = antidiabetic medication, HCC = hepatocellular carcinoma, HR = hazard ratio, CI = confidence interval.

**Table 3 biomedicines-12-01654-t003:** Characteristics of patients with localized tumors according to antidiabetic medication and statin use.

	Metformin		Insulin		Other ADM		Statin		
	Yes (%)	No (%)	Yes (%)	No (%)	Yes (%)	No (%)	Yes (%)	No (%)	Total (%)
**Total**	173 (100)	119 (100)	105 (100)	187 (100)	167 (100)	125 (100)	97 (100)	195 (100)	292 (100)
**Sex**									
Male	143 (83)	92 (77)	79 (75)	156 (83)	139 (83)	96 (77)	78 (80)	157 (81)	235 (80)
Female	30 (17)	27 (23)	26 (25)	31 (17)	28 (17)	29 (23)	19 (20)	38 (19)	57 (20)
**Education**									
Basic	56 (32)	28 (24)	34 (32)	50 (27)	51 (31)	33 (26)	26 (27)	58 (30)	84 (29)
Intermediate	62 (36)	49 (41)	39 (37)	72 (39)	65 (39)	46 (37)	48 (49)	63 (32)	111 (38)
Higher	55 (32)	42 (35)	32 (30)	65 (35)	51 (31)	46 (37)	23 (24)	74 (38)	97 (33)
**Follow-up time (years)**									
Median	0.8	0.6	0.6	0.8	0.8	0.7	0.6	0.7	0.7
Interquartile range	0.1–2.9	0.08–2.4	0.06–2.5	0.2–2.7	0.2–2.9	0.08–2.0	0.1–2.5	0.1–2.8	0.1–2.7
**Duration of diabetes (years)**									
Median	9.6	9.5	12.3	7.8	9.8	8.1	10.2	9.3	9.6
Interquartile range	5.3–13.9	6.0–16.0	8.2–16.9	4.6–12.3	6.8–13.9	3.7–15.8	5.9–15.7	5.3–14.2	5.3–14.9
**Hepatitis**									
Yes	12 (7)	7 (6)	9 (9)	10 (5)	8 (5)	11 (9)	8 (8)	11 (6)	19 (7)
No	161 (93)	112 (94)	96 (91)	177 (95)	159 (95)	114 (91)	89 (92)	184 (94)	273 (93)
**Cirrhosis**									
Yes	41 (24)	31 (26)	29 (28)	43 (23)	43 (26)	29 (23)	22 (23)	50 (26)	72 (25)
No	132 (76)	88 (74)	76 (72)	144 (77)	124 (74)	96 (77)	75 (77)	145 (74)	220 (75)
**Age at HCC diagnosis (years)**									
Median	71	72	70	72	72	70	73	70	71
Interquartile range	65–77	64–80	63–77	65–79	66–79	63–77	66–79	63–78	65–78

ADM = antidiabetic medication, HCC = hepatocellular carcinoma.

**Table 4 biomedicines-12-01654-t004:** Estimation results from Cox proportional hazard models of mortality from all causes, HCC, and other causes in the local subgroup adjusted for sex, education, duration of diabetes and the presence of hepatitis and cirrhosis, as well as year of and age at HCC diagnosis and HCC stage.

	Overall Mortality			HCC Mortality			Other-Cause Mortality		
	HR	95% CI	*n*	HR	95% CI	*n*	HR	95% CI	*n*
**Prediagnostic metformin use**									
Yes	0.96	(0.74–1.24)	159	1.03	(0.77–1.39)	125	0.78	(0.46–1.31)	34
No	1.00	Ref.	110	1.00	Ref.	82	1.00	Ref.	28
**Prediagnostic other ADM use**									
Yes	0.84	(0.63–1.11)	157	0.75	(0.55–1.04)	115	1.16	(0.63–2.12)	42
No	1.00	Ref.	112	1.00	Ref.	92	1.00	Ref.	20
**Prediagnostic insulin use**									
Yes	0.92	(0.69–1.23)	95	0.85	(0.61–1.18)	72	1.19	(0.66–2.16)	23
No	1.00	Ref.	174	1.00	Ref.	135	1.00	Ref.	39
**Prediagnostic statin use**									
Yes	0.92	(0.69–1.23)	88	0.87	(0.62–1.20)	65	1.03	(0.57–1.85)	23
No	1.00	Ref.	181	1.00	Ref.	142	1.00	Ref.	39

ADM = antidiabetic medication, HCC = hepatocellular carcinoma, HR = hazard ratio, CI = confidence interval.

## Data Availability

The data that support the findings of this study are available from the corresponding author upon reasonable request.

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
