# Peer review of "The Association of Metformin, Other Antidiabetic Medications, and Statins with the Prognosis of Hepatocellular Carcinoma in Patients with Type 2 Diabetes: A Retrospective Cohort Study"

_biomedicines, 2024, doi:10.3390/biomedicines12081654_

Round 1

Reviewer 1 Report

Comments and Suggestions for Authors

The manuscript deals with the retrospective analysis of the possible association of antidiabetic medications and statins with the prognosis of hepatocellular carcinoma in patients with type 2 diabetes. The topic is of interest and can impact in the better understanding of the overall and cause-specific survival in hepatocellular carcinoma patients with type 2 diabetes.

The manuscript is well prepared and logically designed. Results and discussion are meaningful. A good comparison to previous works in the fields with cear justification of the achievements in current work are presented. Conclusions are fully supported with the data obtained.

The manuscript can be accepted to publication after minor revision of technical points listed below.

1. Line 93, the dot is missed at the end of the sentence ....were excluded.

2. Tables 1-, an empty raws between the first and second filled raws (lines 177, 193, 221, 241) to be removed.

3. Figures 2 and 3, the headings of each sub-part "M etfo rm in u se: O verall m o rtality". etc. to be corrected as "Metformin use: Overall mortality", etc. This is probably caused by error occured during plots export.

Author Response

Response to Reviewer 1 Comments

The manuscript deals with the retrospective analysis of the possible association of antidiabetic medications and statins with the prognosis of hepatocellular carcinoma in patients with type 2 diabetes. The topic is of interest and can impact in the better understanding of the overall and cause-specific survival in hepatocellular carcinoma patients with type 2 diabetes.

The manuscript is well prepared and logically designed. Results and discussion are meaningful. A good comparison to previous works in the fields with cear justification of the achievements in current work are presented. Conclusions are fully supported with the data obtained.

Response: We thank for the constructive feedback and useful comments on the manuscript, which we have studied carefully. We have taken these suggestions into account with the best of our ability to improve our manuscript.

We have made some modifications to the abstract and major changes to the discussion, please see the highlighted lines in the manuscript.

The manuscript can be accepted to publication after minor revision of technical points listed below.

  1. Line 93, the dot is missed at the end of the sentence ....were excluded.

Response: Thank you for pointing this out. The dot is now added.

  1. Tables 1-, an empty raws between the first and second filled raws (lines 177, 193, 221, 241) to be removed.

Response: Thank you for your kind notice. The empty rows are no removed.

  1. Figures 2 and 3, the headings of each sub-part "M etfo rm in u se: O verall m o rtality". etc. to be corrected as "Metformin use: Overall mortality", etc. This is probably caused by error occured during plots export.

Response: We have now corrected these figure as suggested.

Reviewer 2 Report

Comments and Suggestions for Authors

In a retrospective cohort study including n=1,383 patients with "type-2 diabetes" and, in addition, "hepatocellular carcinoma" (HCC) were evaluated for their clinical prognosis focussing on the potential effect of a  prediagnostic prescription of metformin and statins.

As a result, the prediagnostic use of metformin was associated with a significantly decreased "all cause mortality", "HCC mortality" and "other cause" mortality. This significant association of prediagnostic metformin therapy with the improvement of prognosis, however, could not be affirmed in patients with localized HCC.

No change in overall mortality of HCC-patients was observed in association with prediagnostic insulin or statin therapy. As cited by the authors similar effects also have been reportes earlier in Caucasian and Asian populations.

General considerations and major comments:

- the reported results are of interest and potentially of clinical relevance.

- The major shortcoming of the manuscript is the failure to present any pathophysiological, biochemical and pharmacological mechanisms on the potential protective effect of metformin in HCC-patients. Is it beliefed to be a specific effect on HCC progression? I yes, which are the potential mechanisms?

- The same is true for "statins". Although we know in detail on how statins may stabilize atherosclerotic plaques and thereby retard progression of atherosclerosis and prevent coronary occlusion, it remains unclear wether there might be an additional specific effect in patients with HCC ??

- To help the reader and increase their interest a flow chart of potential pathophysiological mechanisms and pharmacological interactions should be provided

- There may be a considerable amount of confounders that influence the present results. However, it seems that only a limited number of such confounders have been considered in the present study. What do we know on specific cardiovascular risks (e.g. age, presence/absence/severity of coronary artery disease? left ventricular function ?? renal function ? severity of diabetes ?, hypertension etc. ??)

- There are no gender specific evaluations and considerations

Further comments:

- Abbreviations should be listed at the end of the text

- Materials and methods: It is unclear how the data of the various registries considered have been integrated, evaluated and finally included into the study. All included registries should be presented in a table (or tables) within the main text or within an amendment ("supplemental material") including the predefined population, the controls, the specific therapeutic interventions evaluated and the outcomes. Within this context please note that the flow chart in Figure 1 only reflects the baseline and a "rough" selection process.

- Table 1: Define "education"

- Follow-up time (FU): FU is short in Table 1 but extended to 5 years in Fig 2. Please comment on this. We need a time scale that covers the whole history of all patients, for example starting with the first diagnosis of diabetes and/or cardiovascular diagnoses including the specific treatment with metformin and other "anti-diabetics", as well as treatment with statins. In which clinical state of diabetes and of cardiovascular disease has HCC been diagnosed for the first time?

- Table 1: "hepatitis" should be defined in some more details

- Discussion, lines 274-279: the therapeutic "antitumor" mechanisms of metformin should be presented in a flow chart or picture

- Discussion, line 303: the degree of hypercholesterinemia and the degree of atherosclerosis should be presented in more detail as both parameters significantly affect prognosis

- Discussion, line 320: what do the authors understand by "high quality registries" ? How has this term be defined?

- Discussion, lines 327 and upon: the question arises wether the potential prognostic effects of metformin and / or statins do not have anything to do with HCC rather than with a long-term reduction of cardiovascular risk factors thereby including some "better prognosis" of HCC patients. The authors might comment on this.

Comments on the Quality of English Language

Apart from minor corrections the quality of the English language is fine.

Author Response

Response to Reviewer 2 Comments

In a retrospective cohort study including n=1,383 patients with "type-2 diabetes" and, in addition, "hepatocellular carcinoma" (HCC) were evaluated for their clinical prognosis focussing on the potential effect of a  prediagnostic prescription of metformin and statins.

As a result, the prediagnostic use of metformin was associated with a significantly decreased "all cause mortality", "HCC mortality" and "other cause" mortality. This significant association of prediagnostic metformin therapy with the improvement of prognosis, however, could not be affirmed in patients with localized HCC.

No change in overall mortality of HCC-patients was observed in association with prediagnostic insulin or statin therapy. As cited by the authors similar effects also have been reportes earlier in Caucasian and Asian populations.

Response: Thank you very much for taking the time to review this manuscript. Please find the detailed responses below and the corrections highlighted in the re-submitted files.

General considerations and major comments:

- the reported results are of interest and potentially of clinical relevance.

Response: Thank you for this kind comment.

- The major shortcoming of the manuscript is the failure to present any pathophysiological, biochemical and pharmacological mechanisms on the potential protective effect of metformin in HCC-patients. Is it beliefed to be a specific effect on HCC progression? I yes, which are the potential mechanisms?

Response: We thank the reviewer for important feedback. We have now added Figure 4 to illustrate the potential antitumoral mechanisms of metformin.

- The same is true for "statins". Although we know in detail on how statins may stabilize atherosclerotic plaques and thereby retard progression of atherosclerosis and prevent coronary occlusion, it remains unclear wether there might be an additional specific effect in patients with HCC ??

Response: We have now added Figure 5 to demonstrate the effect of statins in HCC.

- To help the reader and increase their interest a flow chart of potential pathophysiological mechanisms and pharmacological interactions should be provided

Response: We have now added Figures 4 and 5 to the manuscript and hopefully, these will clarify pathophysiological mechanisms.

- There may be a considerable amount of confounders that influence the present results. However, it seems that only a limited number of such confounders have been considered in the present study. What do we know on specific cardiovascular risks (e.g. age, presence/absence/severity of coronary artery disease? left ventricular function ?? renal function ? severity of diabetes ?, hypertension etc. ??)

Response: This is an important issue. However, we only had register information available and, unfortunately, register data lack information on severity of coronary artery disease and hypertension as well as left ventricular function. Also, laboratory values were not available. However, insulin use and duration of diabetes is considered to be surrogate marker for severity of diabetes. We have now added these limitations to the manuscript, please see the lines 344-347.

- There are no gender specific evaluations and considerations

Response: Unfortunately, due to strict time limit for resubmission, we are not able to perform gender specific analysis. However, the vast majority of study population is men (80 %), and this applies to all medication groups. In the whole cohort, overall survival (HR 1.13, 95% CI 0.98-1.30) and HCC survival (HR 1.21, 95% CI 1.04-1.41) in women were similar compared with men.

Further comments:

- Abbreviations should be listed at the end of the text

Response: Thank you for pointing this out. Abbreviations are now added.

- Materials and methods: It is unclear how the data of the various registries considered have been integrated, evaluated and finally included into the study. All included registries should be presented in a table (or tables) within the main text or within an amendment ("supplemental material") including the predefined population, the controls, the specific therapeutic interventions evaluated and the outcomes. Within this context please note that the flow chart in Figure 1 only reflects the baseline and a "rough" selection process.

Response: The FinDM database combines several Finnish national registers to identify all persons who have been diagnosed for diabetes and collect their follow-up data. We have now added a Table S1 to demonstrate the used registers.

- Table 1: Define "education"

Response: We thank the reviewer for pointing this out. We have now added a paragraph which defines different education levels. Please see the lines 101-108.

- Follow-up time (FU): FU is short in Table 1 but extended to 5 years in Fig 2. Please comment on this. We need a time scale that covers the whole history of all patients, for example starting with the first diagnosis of diabetes and/or cardiovascular diagnoses including the specific treatment with metformin and other "anti-diabetics", as well as treatment with statins. In which clinical state of diabetes and of cardiovascular disease has HCC been diagnosed for the first time?

Response: The median follow-up time in the whole study population is 0.4 years (interquartile range 0.08 – 1.3) is found in the table 1. In the figure 2, follow-up time is five years because few patients live significantly longer. The main focus of the study was prognosis of HCC and therefore we have decided to begin our follow-up from the diagnosis of HCC rather than the diagnosis of diabetes or cardiovascular diseases. However, duration of diabetes is found in the table 1. Unfortunately, the duration of cardiovascular diseases is not known. We have now added this to our limitations, please see the lines 344-347 and 370-373.

- Table 1: "hepatitis" should be defined in some more details

Response: Presence of hepatitis is based on the register data. We have used International Classification of Diseases versions 8, 9 and 10 coding which are shown in Table S2. We have now added this information to the manuscript, please see the lines 109-110.

- Discussion, lines 274-279: the therapeutic "antitumor" mechanisms of metformin should be presented in a flow chart or picture

Response: We have now added Figure 4 to present antitumoral mechanisms of metformin.

- Discussion, line 303: the degree of hypercholesterinemia and the degree of atherosclerosis should be presented in more detail as both parameters significantly affect prognosis

Response: We agree with the reviewer that the degree of hypercholesterinemia atherosclerosis would be valuable to the manuscript but, unfortunately, the registers lack information on this. We have now added these to the limitations of the manuscript, please see the lines 344-347.

- Discussion, line 320: what do the authors understand by "high quality registries" ? How has this term be defined?

Response: We thank the reviewer for the comment. Data quality is generally considered to be high in Finnish national registers, such as the Hospital Discharge Register (Sund 2012) and the Finnish Cancer Registry. The Finnish Cancer Registry is known to be of high quality in terms of completeness, as 93% of cancer cases have been microscopically verified (Pukkala et al. 2018). Also, all forms of ADM and statins are prescribed by physicians and are not available as over-the-counter medication. ADM attracts more than the basic reimbursement, i.e. these drugs are reimbursed under Special Refund terms and therefore purchases of ADM are particularly accurately recorded. Furthermore, the coverage in the Prescription Register of reimbursed medications prescribed by physicians is virtually complete for study period (Sund, Gissler, Hakulinen, & Rosén, 2014). We have now added some definitions of high-quality register into the manuscript, please see the lines 333-336.

- Discussion, lines 327 and upon: the question arises whether the potential prognostic effects of metformin and / or statins do not have anything to do with HCC rather than with a long-term reduction of cardiovascular risk factors thereby including some "better prognosis" of HCC patients. The authors might comment on this.

Response: We agree with the fact that challenges of confounding by indication are present in observational studies which contain endpoints that have not yet been studied in randomized controlled trials. As various types of medication are initiated to treat conditions other than the one in the focus of an observational study, differences in participants can have an impact on the results. Severity of cardiovascular diseases are not available in our register data and prognostic effect of metformin to overall mortality can be partly caused by reduction of cardiovascular risk factors. However, the prediagnostic metformin use was associated with slightly decreased HCC specific mortality which is not unlikely explained by metformin’s cardiovascular effects, please see the lines 370-373.

Reviewer 3 Report

Comments and Suggestions for Authors

This is a retrospective study in which the statistical analysis is critical for interpretation of the data.  The authors adequately explained the methodology of their statistics and the tables are quite clear.  They discussed the strengths and weaknesses of their study and outlined the limitations.  Only minor grammatical corrections are needed in the body of text and there are a few missing words (example is line 53).  

References and disclaimers are suitable.

Concern:  One thing that the authors may want to address is that diabetes affects men and women differently due to metabolic flux.  20% of the subjects are women.  How does the data look when the women were eliminated from the study?  How do women fare independent of men?  This may have a more profound effect of metformin statistically.

Comments on the Quality of English Language

English quality is good and only minor word smithing and corrections need to be made.

Author Response

Response to Reviewer 3 Comments

This is a retrospective study in which the statistical analysis is critical for interpretation of the data.  The authors adequately explained the methodology of their statistics and the tables are quite clear.  They discussed the strengths and weaknesses of their study and outlined the limitations.  Only minor grammatical corrections are needed in the body of text and there are a few missing words (example is line 53). 

Response: Thank you very much for taking the time to review this manuscript. We have now added some missing words, please see the line 53-54.

References and disclaimers are suitable.

Response: Thank you for your kind comment.

Concern:  One thing that the authors may want to address is that diabetes affects men and women differently due to metabolic flux.  20% of the subjects are women.  How does the data look when the women were eliminated from the study?  How do women fare independent of men?  This may have a more profound effect of metformin statistically.

Response: Unfortunately, due to strict time limit for resubmission, we are not able to perform gender specific analysis. However, the vast majority of study population is men (80 %), and this applies to all medication groups. There were 152 women in metformin users compared to 122 women in metformin non-user group. In the whole cohort, overall survival (HR 1.13, 95% CI 0.98-1.30) and HCC survival (HR 1.21, 95% CI 1.04-1.41) in women were similar compared with men.

Comments on the Quality of English Language

English quality is good and only minor word smithing and corrections need to be made.

Response: Thank you for your kind comment.

Round 2

Reviewer 2 Report

Comments and Suggestions for Authors

Dear Authors,

with the revised manuscript you have addressed all comments and suggestions of the reviewer. Accordingly the manuscript has significantly been improved. Moreover, the results of the present study are of potential clinical interest and revelance. 

Bernhard Rauch